# Area-Selective, In-Situ Growth of Pd-Modified ZnO Nanowires on MEMS Hydrogen Sensors

**DOI:** 10.3390/nano12061001

**Published:** 2022-03-18

**Authors:** Jiahao Hu, Tao Zhang, Ying Chen, Pengcheng Xu, Dan Zheng, Xinxin Li

**Affiliations:** 1School of Chemical and Environmental Engineering, Shanghai Institute of Technology, Shanghai 201418, China; jiahaobang1996@163.com (J.H.); 206061243@mail.sit.edu.cn (T.Z.); 2State Key Lab of Transducer Technology, Shanghai Institute of Microsystem and Information Technology, Chinese Academy of Sciences, Shanghai 200050, China; xpc@mail.sim.ac.cn (P.X.); xxli@mail.sim.ac.cn (X.L.); 3University of Chinese Academy of Sciences, Beijing 100049, China

**Keywords:** in-situ growth, area-selective, MEMS, H_2_ sensor, micro-hotplate

## Abstract

Nanomaterials are widely utilized as sensing materials in semiconductor gas sensors. As sensor sizes continue to shrink, it becomes increasingly challenging to construct micro-scale sensing materials on a micro-sensor with good uniformity and stability. Therefore, in-situ growth with a desired pattern in the tiny sensing area of a microsensor is highly demanded. In this work, we combine area-selective seed layer formation and hydrothermal growth for the in-situ growth of ZnO nanowires (NWs) on Micro-electromechanical Systems (MEMS)-based micro-hotplate gas sensors. The results show that the ZnO NWs are densely grown in the sensing area. With Pd nano-particles’ modification of the ZnO NWs, the sensor is used for hydrogen (H_2_) detection. The sensors with Pd-ZnO NWs show good repeatability as well as a reversible and uniform response to 2.5 ppm–200 ppm H_2_. Our approach offers a technical route for designing various kinds of gas sensors.

## 1. Introduction

Nanomaterials have been widely studied in semiconductor gas sensors as sensing materials owing to their good properties, such as a large surface area, high aspect ratio, and nano-size effect [1,2,3,4]. With extensive applications in environmental monitoring, the Internet of Things, etc., MEMS-based semiconductor gas sensors became the development trend because of the advantages of their small volume, low power consumption, and easy integration [5,6,7,8]. However, as the sensor size shrinks to a micro-/sub-micro scale, the difficulty in constructing gas-sensing materials on micro-sensors significantly increases, especially for micro-scale MEMS sensors [9,10]. Moreover, the uniformity and stability of the constructed sensing materials on the micro-sensors also need to be paid attention to, as the long-term use of sensors is important.

At present, the patterning of nanomaterials on MEMS gas sensors still suffers from incompatibility between “bottom-up” synthesis processes and “top-down” microfabrication technology [11,12]. Some technologies, such as hard-mask deposition [13,14,15], screen printing [16,17], photolithography patterning [18], nanoimprinting [19], micropen direct writing [20], and hydrophobic/hydrophilic surface treatments coating [21] have been developed for the area-selective construction of nanomaterials. However, these methods still show limitations when exploited for micro-gas sensors. For example, screen printing technology requires a sufficient stiffness of the substrate during the print process, but micro-structures such as suspended micro-hotplates could not bear the pressure of printing. On the other hand, the accuracy of a typical screen printing machine is around a hundred micrometers, which is insufficient for the accurate construction of nano-materials on a micro-hotplate. As for the hard-mask deposition and photolithography patterning method, it needs a flat, enclosed membrane in order to proceed with the patterning process; otherwise, the materials could be deposited at the bottom of the groove, resulting in a non-selective deposition [22]. However, plenty of micro-sensors use a suspended plate structure with opening shapes, which are not compatible with the deposition and patterning process [22,23].

To ensure sensing material immobilization, the typical ceramic-based sensor uses a sintering process at a temperature above 500 °C to form a stable contact between the material and substrate [24,25]. However, this temperature is too high for some nanomaterials to keep their morphology [26,27]. Without this sintering process, sensing materials could crack and fall off during high-temperature working conditions. So far, a direct, in-situ construction of sensing material on the sensors is a good solution to the aforementioned problem. Therefore, it is of great importance to develop an in-situ growth process for nanomaterials on MEMS gas sensors, with a good uniformity and maintained material quality.

Hydrogen (H_2_) is an important alternative source of energy, and it can become dangerous when leakage or indoor accumulation occurs [28]. In recent years, handheld hydrogen detectors have been widely needed in hydrogen refueling stations, and the MEMS gas sensor is a good candidate for on-site H_2_ detection [29,30,31], thanks to the advantages of a small volume, quick response, and low cost. Among H_2_ sensing materials, ZnO nanostructures such as nanorods, nanowires, and nanotubes has been found useful owing to their good properties, such as a large surface area and high aspect ratio [32]. However, their selectivity is limited. Since the interaction of Palladium (Pd) and H_2_ was extensively studied decades ago, the Pd-nanoparticle has been a widely used catalyst for improving selectivity [33,34]. Thus, the use of Pd-modified ZnO NWs in order to develop an H_2_ sensor with good sensitivity and selectivity constitutes a nice choice. Meanwhile, the construction of Pd-modified ZnO nanostructures in the sensing area on MEMS sensors is still a challenge for most microsensors.

In this work, we combine seed layer patterning and area-selective hydrothermal growth to achieve the in-situ growth of ZnO nanowires (NWs) on MEMS micro-hotplate gas sensors. Using this newly developed technology, Pd-modified ZnO NWs are densely grown on the sensing area of the sensor chips. We characterize the Pd-ZnO NWs and evaluate the H_2_ sensing performance of the micro-hotplate sensors, such as the sensitivity, repeatability, and uniformity.

## 2. Materials and Methods

### 2.1. Chemicals

Chemicals including zinc acetate (Zn(CH_3_COO)_2_), zinc nitrate (Zn(NO_3_)_2_), palladium chloride (PdCl_2_), hexamethylenetetramine, sodium tetrachloropalladate(II) (Na_2_PdCl_4_), potassium bromide (KBr), poly(vinyl pyrrolidone) (PVP, Mw ≈ 40,000), and L-ascorbic acid (AA) are purchased from Sigma-Aldrich (Merck Limited, Shanghai, China). Photo AZ4620 is purchased from Microchemicals Co. Ltd. (Versum Materials (Shanghai) Co., Ltd. Shanghai, China). Absolute ethanol is purchased from Shanghai Lingfeng Chemical Reagent Co. Ltd. (Shanghai, China). PRS-3000 photoresist stripper is purchased from Fanmeng new material Shanghai Co. Ltd. (Shanghai, China).

### 2.2. Fabrication of Micro-Hotplate with Patterned Hydrophobic Layer

The micro-hotplate is comprised of a Si_3_N_4_ supporting layer, Pt heater, comb electrodes, isolating layer, and hydrophobic heptadecafluorodecyl-trimethoxysilane (FAS-17) layer. The fabrication process of micro-sensor chips can be found in our previous work [28,35]. The FAS-17 layer is grown by molecular vapor deposition (MVD, Applied Microstructures, MVD-100E) [36]. After the wafer is cut by laser dicing, each chip is mounted on a ceramic package and wire bonded. The Pt thin film is used as the heater to provide the desired temperature for gas sensing. With patterned FAS-17 SAM, most of the surface on the sensor chip is hydrophobic, while the sensing area is hydrophilic.

### 2.3. In-Situ Preparation of Pd-Modified ZnO Nanowires on Sensor Chip

The Pd-modified ZnO nanowires are prepared in three steps. Firstly, the ZnO seed layer is loaded and sintered. Then, ZnO NWs are grown in situ using a hydrothermal method. Thirdly, Pd nanoparticles are synthesized on the ZnO NWs. The process is detailed as follows: (i) ZnO nano-crystal seed solution is prepared by dissolving 22 mg Zn(CH_3_COO)_2_ into 20 mL ethanol under ultrasound conditions. Then, the sensor is dip-coated in the aforementioned solution three times. With the help of the hydrophobic SAM layer (schemed as a green layer on the surface), the solution remains in the sensing area of the chip. The chip is further dried in an 80 °C oven for 30 min and sintered at 350 °C for 20 min in a tubular furnace to obtain a seed layer of ZnO. (ii) The hydrothermal-growth compatible photoresist (PR) AZ4620 is coated on the frame and supporting beams using a nano-inkjet printer (Sonoplot Miroplotter II) and is fully dried in an oven. Meanwhile, 0.17 g of hexamethylenetetramine and 0.33 g of Zn(NO_3_)_2_ are dissolved into 100 mL of deionized water as a stock solution. Thereafter, the ZnO-seed coated micro-chip is carefully flipped and floated on the stock solution, and ZnO NWs can be grown in situ onto the sensing area of the chip by using a hydrothermal method [37]. The hydrothermal process is conducted at 90 °C for 4 h. Then, the sensing chip with ZnO NWs is taken out of the solution, and the photoresist is removed by a PRS-3000 remover. Finally, the chip is washed with deionized water three times and dried. (iii) The synthesis steps of Pd nanoparticles are as follows [38]: 105 mg of PVP is added to 11 mL of water, and 60 mg of AA, 600 mg of KBr, and 57 mg of sodium tetrachloropalladate(II) (Na_2_PdCl_4_) are added, stirred for 3 h under oil bath heating at 80 °C and cooled to room temperature. The product is collected by centrifugation and washed with ethanol and water. Centrifugation and washing are repeated several times to remove excess bromine ions. The final product is dispersed in 8 mL of deionized water. After the synthesis process, the sensor chip with in-situ-grown ZnO NWs is immersed into the diluted Pd dispersion. The Pd nanoparticle is immobilized onto ZnO NWs after ultrasonic treatment for 5 s. Finally, the chip is rinsed with ethanol and deionized water, and dried. After the growth process, the sensors are aged at 250 °C for five days before the H_2_ sensing experiments.

### 2.4. Material Characterization

ZnO is characterized by scanning electron microscopy (SEM, HITACHI S4800, Tokyo, Japan) working at 3–5 kV, and equipped with an energy dispersive spectroscope (EDS, Aztec X-Max 80, Oxford Instruments, Oxford, UK). A Tecnai G2TF20S-TWIN microscope (Thermo Fisher Scientific Inc., Hillsboro, OR, USA) with an accelerating voltage of 200 kV is used to obtain the transmission electron microscopy (TEM) images of ZnO and metal nanoparticles. The crystal structures of the samples are analyzed by X-ray diffraction (XRD, Bruker model D8 focus diffractometer, Bruker Corporation, Ettlingen, Germany) that is equipped with a Cu anode to produce an X-ray (40 kV, 40 mA). The XRD patterns of the sample are collected in a continuous scan mode from 10° to 90°.

### 2.5. Temperature Calibration and Gas Sensing Measurement Setup

The temperature of the micro-hotplate is calibrated by an infrared imager (Fluke TiX 560, Everett, WA, USA) with a macro lens (FLK-LENS/25MAC2, Everett, WA, USA). This equipment is capable of measuring the temperature down to a size of 25 μm. H_2_ with desired concentrations are generated by diluting the pure H_2_ gas with clear air. Standard H_2_ gas with a concentration of 99.99% is purchased from Weichuang Gas Corporation, Shanghai, China. The heating voltage of the micro-hotplate sensor is supplied by a DC power source (Agilent E3632, Santa Clara, CA, USA). The resistance values of the sensors are recorded with a multimeter (Agilent 34401A, Santa Clara, CA, USA). The sensor is placed into a testing chamber with quartz observing windows. The testing chamber is connected to a gas generator, which can provide H_2_ gases with desired concentrations. The sensor response to H_2_ gas is defined as R_a_/R_g_, where R_a_ is the resistance of the sensor in the clean air and R_g_ is the resistance in the H_2_ gas.

## 3. Results and Discussions

### 3.1. Micro-Hotplate Sensors

The SEM image and schematic of the micro-hotplate sensor are shown in Figure 1a,b: the suspended hotplate has a diameter of 300 μm and is connected to four narrow supporting beams that are clamped to the substrate. The ring-shaped Pt heaters are located on the rim of the circular hotplate and covered by insulating layer. The inner part of the plate is patterned with dense comb electrodes. The ring-shaped Pt heaters are vertically insulated from the sensing signal measurement circuit. Thanks to the optimized design, the sensor power consumption is only 13 mW at 150 °C.

### 3.2. In-Situ Growth of Pd-Modified ZnO NWs on Micro-Hotplate Chip

The area-selective growth process of Pd-modified ZnO NWs is depicted in Figure 1c–h. The sensor chip after dip-coating in the seed layer solution is shown in Figure 2a. The solution remains only in the sensing area of the micro-hotplate thanks to the hydrophilic treatment. The contact angle is about 112° according to our previous study [28,36]. The hydrophobic SAMs could also be potentially exploited for the accurate loading of other kinds of solutions. After the chip is sintered in air, the seeds for ZnO NWs growth are formed in the sensing area. As the high-temperature sintering process removes the SAM on the surface [36], it is essential to use photoresist as a compatible barrier layer for area-selective hydrothermal growth, as illustrated in Figure 1e,f. The results of hydrothermal growth are shown in Figure 2b–f. The ZnO NWs are densely, uniformly grown in the sensing area of the micro-hotplate with a diameter of 80 ± 5 nm. The boundary of the ZnO NWs is quite clear, with dense NWs grown within the sensing area and barely shown outside of the sensing area, as shown in Figure 2b–d. The result indicates that the photoresist can act as the barrier layer in order to confine the ZnO NWs growth within the sensing area. The whole in-situ growth process has been successfully repeated tens of times, which indicates that our method is highly reproducible. The in-situ growth results also demonstrate that our method features a good area selectivity and good compatibility, and that there is the potential for the in-situ construction of other kinds of nanomaterials on sensors by the hydrothermal method.

### 3.3. Characterization of Pd-Modified ZnO Nanowires

The crystal structures of the ZnO NWs and Pd nanoparticles are identified by XRD. As shown in Figure 3a, all the diffraction peaks are well matched with the standard ZnO (JCPDS card number: 79-0208). The main diffraction peaks located at 2θ = 36.1°, 31.6° and 34.3° could be assigned to the (101), (100) and (002) planes of ZnO, respectively [39]. The XRD pattern in Figure 3b shows peaks located at 40.1°, 82.1° and 46.7°, which can be assigned to Pd (JCPDS card number: 05-0681). The TEM image in Figure 3c shows that the Pd has been decorated on the ZnO NWs and that the diameter of the one-dimensional nanowires measures within 75–85 nm. The TEM image with a high magnification in Figure 3d indicates that the Pd nanoparticles have a cubic shape with a length scale that ranges from 20 to 30 nm.

### 3.4. Gas Sensing Performance

We evaluate the H_2_ sensing performance of the micro-hotplate sensors with in-situ-grown Pd-modified ZnO NWs. Figure 4a shows the resistance changes of the sensor in response to 100 ppm H_2_ at different working temperatures of 50–300 °C. The maximum is achieved at 150 °C, and this temperature is set as the optimal working temperature in the following experiments. Figure 4b shows a gas sensor response after being sequentially exposed to H_2_ with concentrations from 2.5 to 200 ppm, and the resistance change increases with the H_2_ concentration. At 150 °C, the average response time of the sensor is about 2.5 min, and the recovery time is about 2.7 min. Repeatability is evaluated by exposing the sensor to 50 ppm H_2_ three times, as shown in Figure 4c. For the reproducibility, the growth process is repeated on six sensors, and these sensors are tested in 100 ppm H_2_, as shown in Figure 4d. The results indicate a good reproducibility of our method. To investigate the selectivity of the Pd-modified ZnO NWs sensor, five commonly existing gases, acetone, toluene, ammonia, methane, and ethanol, with concentrations of 100 ppm are selected as interfering gases. Our H_2_ sensor shows negligible responses to the interfering gases, as shown in Figure 4e. The result indicates a good selectivity of the sensor to H_2_ gas. The long-term stability of the sensor is investigated by monitoring the response of the same sensor to 100 ppm H_2_ detection for 30 days. As shown in Figure 4f, the sensor response after 30 days is comparable to that measured on the first day (the response only decreases by 3.6%). Thanks to the good long-term repeatability and stability of the Pd-modified ZnO NWs sensor, our method facilitates the development of MEMS sensors for on-site gas detection.

## 4. Conclusions

In summary, Pd-modified ZnO NWs are synthesized in situ in the sensing area of the micro-hotplate sensor. Firstly, the seed layer of ZnO is formed with the help of the hydrophobic SAM, and then the nanowires are grown in situ with a patterned photoresist layer. Afterward, Pd nanocubes are synthesized and modified on the ZnO NWs. The results indicate that the area-selective grown nanowires are dense, uniform, and firm. H_2_ gas sensing experiments are implemented to evaluate the performance of area-selective constructed nanomaterial. The sensor exhibits good responses to H_2_ with a concentration from 2.5 ppm to 200 ppm at the optimal working temperature of 150 °C and shows a satisfactory sensitivity, repeatability, selectivity, and long-term stability. As a result, the method developed in this work features a good compatibility with MEMS sensor fabrication and nanomaterial synthesis, and it has the potential to be used for the direct, in-situ construction of other kinds of nanomaterials on MEMS gas sensors.

## Figures and Tables

**Figure 1 nanomaterials-12-01001-f001:**
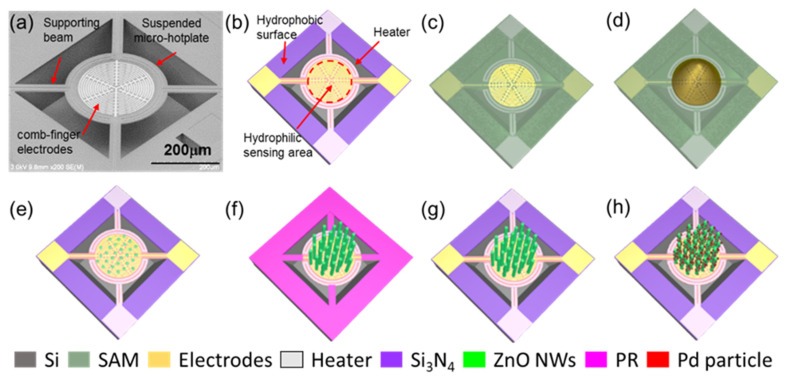
(**a**) SEM image of micro-hotplate and the structure of the sensor. (**b**) Design of micro-hotplate sensor. The sensing area of the hotplate, which is surrounded by the Pt heaters, is hydrophilic, while other parts of the chip are hydrophobic. (**c**–**h**) Process steps of in-situ growth of Pd-modified ZnO NWs on the chip.

**Figure 2 nanomaterials-12-01001-f002:**
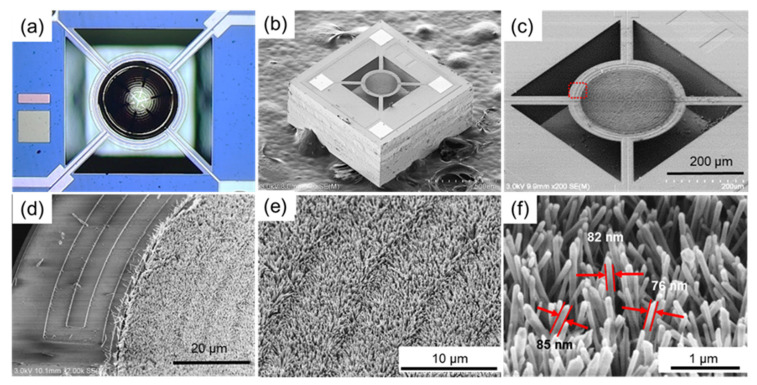
(**a**) An optical image showing the seed layer solution loading process. (**b**,**c**) SEM image of sensor chip after ZnO NWs growth process; the nanowires are immobilized in the sensing area, while NWs are barely found in other areas. The magnified SEM image of (**c**) is shown in (**d**). The boundary of the sensing area is quite clear. (**e**) The ZnO NWs are densely grown in the sensing area. (**f**) Zoom-in view of the dense ZnO NWs; the nanowire diameter is around 80 ± 5 nm.

**Figure 3 nanomaterials-12-01001-f003:**
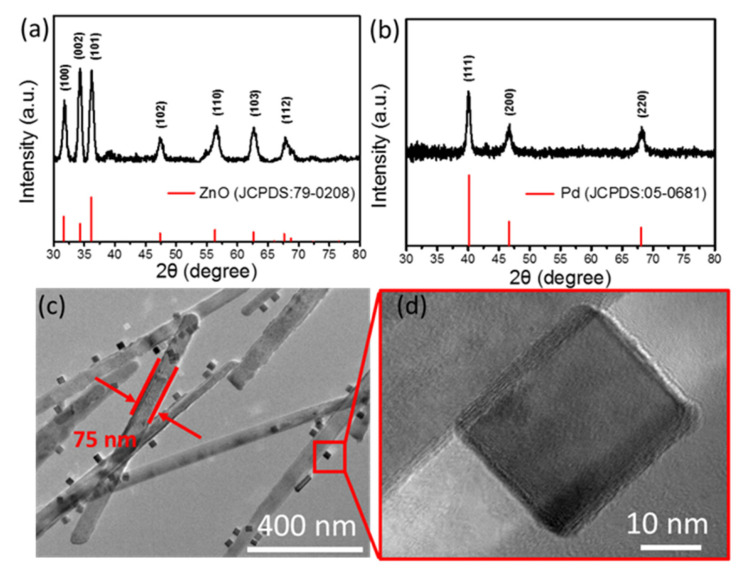
(**a**,**b**) XRD images of ZnO NWs and Pd nanoparticles. (**c**) TEM image of Pd-modified ZnO NWs and (**d**) zoom-in view of the Pd nanoparticle.

**Figure 4 nanomaterials-12-01001-f004:**
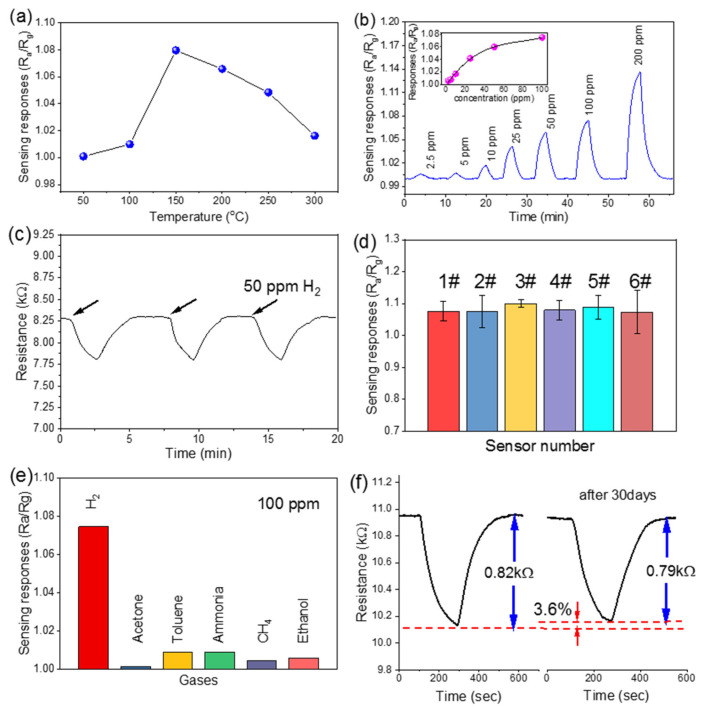
(**a**) Temperature-dependent resistance change of the H_2_ sensor, where the concentration of H_2_ is 100 ppm. (**b**) Sensing responses to H_2_ from 2.5 ppm to 200 ppm. Inset shows the relationship between the resistance change and gas concentration. (**c**) Repeatability test results of the H_2_ sensor to 50 ppm H_2_ at 150 °C. (**d**) Responses of six sensors (with serial numbers from 1# to 6#, prepared from different batches) to 100 ppm H_2_ gas. (**e**) Selectivity of the Pd-modified ZnO NWs sensor to five different gases. The concentration of all interfering gases is 100 ppm. (**f**) Long-term stability test results of the sensor to 100 ppm H_2_ gas. The interval between the two tests is 30 days.

## Data Availability

Not applicable.

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
