# Peer review of "Area-Selective, In-Situ Growth of Pd-Modified ZnO Nanowires on MEMS Hydrogen Sensors"

_nanomaterials, 2022, doi:10.3390/nano12061001_

Round 1

Reviewer 1 Report

Presented manuscript is focused to synthesize Pd-modified ZnO NWs  in the sensing area of the micro-hotplate sensor.

Several remarks on the manuscript:

  1. What are the reasons or the advantages of using nanowires? The authors must demonstrate what is the novelty?
  2. Conclusion part should summarize all the important findings of the manuscript. Please reformulate it.

Reviewer 2 Report

The manuscript is well written there are some issues needs to be addressed before publication

1. In the section 3.4

It is mentioned that  "Repeatability is evaluated by exposing the sensor to 100 ppm H2 three times, as shown in Fig.4 c." but in the caption and figure it is written 50 ppm. Which is the correct one 100ppm or 50 ppm.

2. What is the response time and recovery time for the sensor at 150 deg c. include the same in the manuscript.

3. Reason for choosing Pd  nanoparticles in this work should be elaborated in the Introduction section which is missing.
